# Disrupting the Mood and Obesity Cycle: The Potential Role of Metformin

**Stacey N. Doan** [1,2,*], **Sunita K. Patel** [2] , **Bin Xie** [3], **Rebecca A. Nelson** [4] **and Lisa D. Yee** [5,*]

1   Department of Psychological Science, Claremont McKenna College, Claremont, CA 91711, USA
2   Department of Population Sciences, City of Hope, Duarte, CA 91010, USA
3   School of Community & Global Health, Claremont Graduate University, Claremont, CA 91711, USA
4   Department of Computational and Quantitative Medicine, City of Hope, Duarte, CA 91010, USA
5   Department of Surgery, City of Hope, Duarte, CA 91010, USA
*   Correspondence: staceyndoan@gmail.com (S.N.D.); lyee@coh.org (L.D.Y.)

**Abstract:** Mounting evidence links obesity, metabolic dysfunction, mood, and cognition. Compromised metabolic health and psychological functioning worsen clinical outcomes, diminish quality of life, and contribute to comorbid conditions. As a medication with both insulin-sensitizing and anti-inflammatory effects, metformin affords the exciting opportunity to abrogate the bidirectional relationship between poor metabolic health and psychological function. In the current paper, we review the literature linking metformin to mood and cognitive function, examine potential underlying mechanisms, and suggest new directions for investigating the role of metformin in increasing adherence to health behavior recommendations.

**Keywords:** obesity; metabolic dysfunction; mood; cognitive function; metformin; breast cancer risk

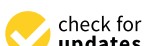



## 1. Introduction

With the burgeoning rise in obesity in the United States, an increasing number of Americans will face the deleterious health effects of excess weight and adiposity including Type 2 diabetes mellitus (T2DM), cardiovascular disease, and malignancies including colorectal, endometrial, and postmenopausal breast cancer [1,2]. Based on data from the National Health and Nutrition Examination Surveys (NHANES) from 1999–2000 through 2017–2018, the age-adjusted prevalence of obesity in U.S. adults increased from 30.5% to 42.4%; similarly, the age-adjusted prevalence of severe obesity rose from 4.7% to 9.2% [3]. For 2017–2018, middle-aged adults (40–59 years) had the highest prevalence of obesity at 44.8%; women had a higher prevalence of severe obesity (BMI $\geq$ 40 kg/m$^2$) compared to men, at 11.5% versus 6.9% [3]. The prevalence of obesity was highest in non-Hispanic Black adults at 49.6%, followed by Hispanic (44.8%) and non-Hispanic White (42.2%) adults [3]. The steady rise in obesity over the past two decades is an urgent call to action.

Current strategies for weight loss include recommendations for healthy eating and regular exercise. While healthy eating and active living/exercise are effective strategies for managing obesity [4], engaging in healthy dietary behaviors and sustained physical activity is notoriously difficult. Although people who are overweight or have obesity are able to lose weight by short-term reports [5], NHANES data also indicate that most individuals are unable to maintain weight loss [6], and fewer adults who are overweight or have obesity report trying to lose weight [7]. This lack of adherence to a healthy lifestyle is due to multiple reasons, including (1) low tolerance to high-intensity exercise and higher perceived exertion in overweight individuals [8–10], and (2) difficulty in initiating and sustaining healthy lifestyle changes due to emotional and cognitive dysregulation integrally linked to obesity and overweight status [11,12]. Therefore, for people who are overweight or suffer from obesity, standard of care recommendations for lifestyle changes alone are likely to have little impact on reducing obesity risk. The difficulties encountered by many

adults to reverse weight gain via recommended lifestyle practices of exercise and caloric restraint, despite the compelling incentive of health-related benefits, indicates a more complex problem.

In the current paper, we provide an overview of the literature demonstrating the association between obesity, metabolic dysfunction, mood, and cognitive dysregulation. We then describe the common underlying pathophysiology. In the final section of the paper, we review animal and human models demonstrating the potential role for metformin, an oral hypoglycemic drug, in disrupting the intransigent cycles of metabolic, mood, and cognitive dysfunction. We end with a discussion and recommendations for future research.

## 2. Obesity and Metabolic Dysfunction

The chronic imbalance of energy intake versus energy expenditure leads to excessive fat accumulation and obesity. Obesity often leads to insulin resistance and changes in fuel utilization between carbohydrates, lipids, and protein, whereby insulin-stimulated glucose uptake by muscle and adipose tissue is impaired. Hyperinsulinemia ensues when cells become unresponsive to insulin, which fosters a proinflammatory milieu in adipose tissue with ectopic fat storage and aberrant energy usage [13,14]. Excess adiposity, especially visceral adiposity, also promotes chronic low-grade inflammation by macrophage, adipocyte–preadipocyte production of proinflammatory cytokines such as C-reactive protein (CRP), interleukin 6 (IL-6), and tumor necrosis factor alpha (TNF$\alpha$) and adipokines such as leptin [15,16]; these endocrine effects of adipose tissue inflammation are considered causative of systemic inflammatory pathway activation and lead to insulin resistance. The co-occurrence of obesity/visceral adiposity, insulin resistance, dyslipidemia, and hypertension comprises metabolic syndrome [17,18], which carries increased risk for Type 2 diabetes mellitus (T2DM), cardiovascular disease (CVD) [19], and postmenopausal breast cancer [20–22]. Obesity, proinflammatory signal transduction, and insulin resistance form a vicious cycle of dysregulated metabolism with deleterious health effects.

## 3. Relations between Mood Dysregulation and Metabolic Dysfunction

Obesity and overweight status are closely associated with mood dysregulation in a codependent, bidirectional manner [23,24]. Obesity leads to a 25% increase in odds of mood and anxiety disorders [25]. The association between depression and obesity is more consistently evident among women than men [26]. Even after controlling for race and socioeconomic variables, young women with overweight status or obesity are more likely to report having sustained depressive mood than women who are lean [27]. Similarly, depression, as assessed by the Major Depression Inventory, is associated with increased risk of T2DM, as observed in population-based research [28]. Notably, depression is detected even at the prediabetes stage, with data demonstrating a 37–60% increase in prospective risk of developing T2DM among individuals with depression [29].

The relationship between depression and insulin resistance is observed even in individuals without abnormal or excessive fat accumulation. A prospective, longitudinal study showed that children with depression, measured with the Child Depression Inventory, develop insulin resistance independent of changes in BMI [30]. Consistent with these data, a recent meta-analysis suggests a small but significant association between depression and insulin resistance [31]. Note that meta-analyses showing an association between depressive symptoms or general distress and T2DM appear robust with both diagnostic (e.g., clinical records) and nondiagnostic (Centers for Epidemiologic Studies for Depression Scale, General Health Questionnaire) measures of depression [29]. Relatedly, even general negative affect, rather than diagnosed depression, is associated with metabolic health [32]. Higher negative affect or emotional distress (e.g., anxiety, depression, stress, sadness) [33] and lower positive affect or pleasant feelings or emotions (e.g., joy, calmness, interest, enthusiasm) [34], as assessed using the Positive and Negative Affect Schedule, are associated with increased BMI; for women, the effect is stronger [34]. Interestingly, the relationship between lower positive affect and higher BMI appeared to be explained by physical ill

health [34]. Mood disorders such as depression are thought to lead to excessive weight gain because individuals with depression tend to have lowered energy and therefore are less physically active [35], and negative affect tends to be associated with higher intake of sweet, high-fat, and energy dense foods [36]; depression may also lead to use of food in an attempt to cope with emotional distress, given that food intake corresponds with acute physiological changes (e.g., increased serotonin) that can alleviate discomfort for the short term [37]. Moreover, analyses suggest that depression was more likely to precede obesity, rather than vice versa [38]. Additionally, pharmaceutical treatments for mood disorders can also induce weight gain [28,39–41], with potential to exacerbate insulin resistance and the inexorable cycle of mood and metabolic dysfunction [42,43].

## 4. Relationship between Cognitive Function and Metabolic Dysfunction

In addition to mood, disruptions in cognition, specifically to higher level executive functioning, are observed among those with obesity [44]. Executive function (EF) is a set of higher-order cognitive processes necessary for goal-directed behavior, including working memory (i.e., short term storage of relevant, immediate information), inhibitory control (i.e., ability to control one's attention, behavior, thoughts, and/or emotions to override impulsive or automatic/conditioned responses), and shifting/flexibility (i.e., ability to adapt behavior and thoughts to new, changing, or unexpected events) [45], as well as decision-making, which includes elements of applying and following rules [46], verbal fluency as a measure of ease or speed of semantic processing [47], and planning (i.e., forethought for future adaptive responses) [48]. EF skills are central for directing and guiding behavior, particularly in situations that are nonroutine or effortful, such as initiating health behavior change [49].

The majority of studies examining the relationship between obesity and EF have focused on differences between individuals with obesity and normal weight, with others focusing on overweight and normal weight individuals [44]. A meta-analysis conducted by Yang et al. [44] shows that a variety of EF tasks have been used to investigate the role of cognitive processes in excessive weight gain; inhibitory control and planning were the most common functions identified. Overall, results indicate EF deficits among overweight individuals or those with obesity. Specifically, individuals with obesity exhibited poor EF across all domains (i.e., working memory, inhibitory control, shifting/flexibility, decision-making, verbal fluency, planning), while overweight individuals only showed significant deficits in inhibitory control and working memory relative to normal weight controls [44]. Age, BMI, and sex did not seem to influence the pattern of results. Regarding working memory and decision-making, the type of task seemed to matter. For memory assessment, the digit span task and the delay discounting or Iowa gambling task exhibited larger effect sizes than the letter-numbering sequencing task. An important limitation identified by this meta-analysis, however, is the small number of studies focusing only on individuals who are overweight (rather than with obesity); lack of consideration of use of antiobesity medication and presence of a psychiatric disorder in many studies were also noted.

Studies examining multiple measures of EF reveal inconsistent results regarding the relationship between EF deficits and excess adiposity/obesity. The Baltimore Longitudinal study of aging reported mixed results regarding type of cognitive task as well as the relationship between measures of adiposity and cognitive outcomes [50]. Study findings suggest that BMI and waist circumference were associated with poorer prospective memory; longitudinally, all three measures of adiposity (BMI, waist circumference, waist–hip ratio, or WHR as ratio of measurements of waist to hip circumference) demonstrated declining performance on the Benton Visual Retention test with increasing body size. While cross-sectional analyses showed that BMI was associated with significantly worse performance on the Letter and Category Fluency test, there were no longitudinal effects on any of the language measures (Letter and Category Fluency and Boston Naming) [50]. Regarding executive function (using the Trail Making B task, thought to be indicative of working memory and shifting abilities) [51], WHR was associated with slower performance over

time. Surprisingly, however, obesity was associated with better attention and visuospatial ability. The relatively well-educated (average >16 years of education), and older (mean age 55.5, SD: 16.9) sample makes the results difficult to generalize to other populations.

Results between adiposity and cognitive function were more consistent in a sample of middle-aged adults. The Framingham Heart study focused on BMI and WHR, both of which are risk factors for cardiovascular disease, in a sample of over 1800 men and women (age 40–69) at baseline. Results suggest that obesity and hypertension were related to worse executive function performance (using the Trails B test) [52]. The authors also found greater age-related cognitive decline in individuals with obesity and suggest the importance of controlling central obesity to reduce age-related cognitive declines. The relationship between BMI and cognition was further explored in the Whitehall II study [53], which assessed BMI over the life course at 25 years of age, during early midlife (mean age = 44), and late midlife (mean age = 61). Cognition was assessed in late midlife using the Mini Mental State Examination (MMSE), a screening measure of global cognition, and tests of short-term memory (a free recall task) and EF (reasoning and verbal fluency). Results suggest a curvilinear relationship, with both individuals who were underweight or with obesity having poorer cognition. Specifically, cumulative obesity (obesity at two or three time points) was associated with lower scores on the MMSE and measures of memory and EF; an increase in BMI from early to late midlife correlated with lower levels of executive function at late midlife [53]. In a review of 88 studies, Favieri et al. confirmed an inverse relation between obesity and a range of executive functioning measures (Wisconsin Card Sorting Test, Trail Making Test, Stroop Color-Word Task, Digit Span Test, Delay Discounting Task) [11].

These data suggest that obesity is a disorder of appetitive motivation, rather than simply a disorder of disruptions in homeostatic mechanisms of food intake. Dysfunction of the central melanocortin system, which is involved in regulation of energy homeostasis, food intake, satiety, and body weight, is also implicated in the pathogenesis of obesity [54,55]. The motivation to consume certain foods activates the mesolimbic dopamine system; cognitive functions (e.g., reward, desire) associated with the mesolimbic pathway are implicated in addiction [56]. At the same time, however, researchers have argued that these models need to incorporate EF processes, namely, those related to inhibitory control, which are localized in the prefrontal cortex [57]. Neurobiological and behavioral evidence suggests that individuals who have lower EF abilities are particularly susceptible to intake of high caloric foods, as well as weight gain [57].

## 5. Mechanisms Linking Metabolic Health, Mood, and Cognitive Functioning

Researchers have theorized that both behavioral and biological mechanisms explain the relationships between mood, cognitive functioning, and metabolic health. From a behavioral perspective, depressed mood/negative affect leads to an increased preference for high-caloric food, which can serve as a form of emotion regulation [58]. Moreover, stress and negative affect can degrade higher executive functioning competencies that are critical to self-control. Finally, from a physiological perspective, long-term negative affect and/or chronic stress can lead to dysregulation in endocrine and immune systems that affect both brain and metabolic health, which can then, in turn, affect cognitive and mood disruptions and higher food intake, leading to a vicious, inescapable cycle.

Observational studies of stress, using both life events as well as subjective distress scores [59], and lab-based manipulations of mood demonstrate that negative cues increase the salience of immediate, concrete goals, leading to preference for indulgent rather than healthy food [60]. Mood dysregulation is associated with poor health behaviors, including physical inactivity [61], binge eating [62], and increased caloric intake [63]. Relatedly, stress/negative affect [64] and depression [65] impair executive functioning abilities which are crucial to initiating and maintaining health behaviors. This is particularly problematic because EF is essential for behaviors (including health behaviors), which have an immediate cost in terms of time and effort but offer health benefits in the long term. Evidence suggests

that the effects of mood/stress on health and health behaviors are mediated through disruptions in executive functioning abilities [66].

Prolonged depression and negative mood states involve activation of the hypothalamic–pituitary–adrenal axis, sympathoadrenal system, and proinflammatory cytokines [67]. Dysregulation in these systems can induce insulin resistance and contribute to diabetes risk [67]. A high-fat diet causes insulin resistance and T2DM by disrupting signaling circuits and neurotransmitter systems in the prefrontal cortex associated with motivation, reward, depression, and anxiety [68]. Insulin resistance is also associated with loss of motivation and heightened food-seeking behaviors thought to be mediated by differences in the anterior cingulate cortex—a known hippocampal motivational network that contributes to both depression and insulin resistance [69]. Evidence suggests that cortisol, the end-product of the hypothalamic–pituitary–adrenal axis (HPA), one of the main stress response systems, is positively associated with weight gain and enhanced secretion of proinflammatory hormones and cytokines (adipokines) by adipose tissue depots [70]. Proinflammatory cytokines, particularly IL-6 and CRP, have been implicated in both insulin resistance and T2DM [16]. Proinflammatory signaling may underlie depression in the setting of obesity and dysfunctional metabolism [71].

Obesity is also linked to mood dysregulation as a risk factor. Chronic inflammation arising from higher fat mass and metabolic dysfunction appears to be associated with psychological effects such as depression and anxiety [72]. In a study of data from a mental health questionnaire of participants in the U.K. Biobank ($n$ = 145,668), Casanova et al. demonstrated that higher adiposity leads to higher odds of depression, severity of depression, and lower wellbeing, regardless of genetic predisposition to metabolic dysfunction (e.g., adiposity genetic variants with favorable or unfavorable metabolic profiles based on HDL cholesterol, triglycerides, and T2DM risk) [73]; limitations of the analysis include lack of diversity in the European study population and potential bias in the subset of participants involved in the mental health questionnaire substudy. Interestingly, the metabolically favorable adiposity variants were associated with higher levels of the proinflammatory cytokine CRP [73].

In sum, increasing evidence suggests the association of obesity and insulin resistance with depressed mood and cognitive dysfunction as an interrelated network that can become intransigent and bidirectionally entrained (Figure 1). Negative affect and stress bias individuals towards emotional responses, while at the same time degrading self-control abilities, leading to higher food consumption [74,75] and increasingly sedentary behaviors [76–78]. These behaviors over time can lead to reduced functioning in various quality of life domains, perpetuating, and possibly worsening, depressed mood. Relatedly, metabolic dysfunction including insulin resistance also appears to be associated with increases in mood disorders, suggesting bidirectional and convergent effects [23]. Higher BMI also decreases the effectiveness of antidepressants [79,80], and depression predicts unfavorable outcomes in a range of weight loss interventions, including surgical [81] and behavioral [82], as well as poor weight loss maintenance [83]. In fact, even nonclinical assessments of mood, such as stress (using the Perceived Stress Scale) [84,85], or primarily nondepressed individuals (BDI scores of <10) [86], show that higher levels predict lower efficacy of interventions, likely due to decreased engagement [84]. Studies also demonstrate that individuals who experience remission from depression are more likely to lose weight from lifestyle interventions than those who do not [86,87], perhaps due to higher levels of engagement in physical activity [88].

These results are of great significance to both clinicians and researchers. As reviewed, chronic stress and mood dysregulation result in increased preference for energy-dense food; consequently, physiological responses to stressors are intertwined with regulation of appetite [89]. Recently, we published a comprehensive model [90] in which we argue that improving cognitive and emotional capacities in at-risk individuals can lead to greater likelihood of adherence to healthy behaviors. Addressing cognitive and emotional barriers to behavior change before implementing a lifestyle intervention may lead to increased

adherence [90,91]. Empirical data demonstrate that improvements in mood are associated with adherence to medical and health behavior recommendations [92,93].

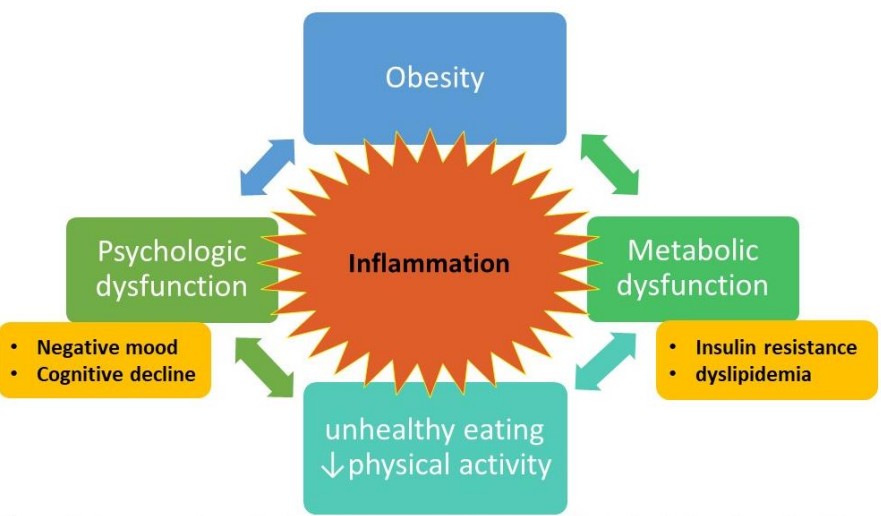

**Figure 1.** Conceptual model of obesity = metabolic and psychological dysfunction = inability to effect behavioral change to reverse obesity. Bidirectional arrows indicate feedback loops that form the intractable cycle. Inflammation is central to the model, with cytokines and other inflammatory factors fueling the bidirectional connections. Figure designed by L.D.Y. and S.D.

As both mood and higher-level cognitive abilities associated with self-control are (1) disrupted in the context of obesity and (2) share interconnected pathophysiology [94,95], interventions that target these factors may be key to improving health behavior uptake and adherence. While behavioral interventions such as cognitive training [96] and mindfulness-based therapy [97] can improve cognition and mood, respectively, such methods are often just as time-intensive as the lifestyle behavior changes themselves; as a result, behavioral methods may prove challenging to implement. Just as importantly, antidepressant use is often associated with weight gain [98] and diabetes risk [99].

## 6. Metformin Provides an Exciting Pharmaceutical Intervention with Potential to Restore Metabolic Health while Simultaneously Improving Mood and Cognition

Metformin reduces insulin resistance associated with obesity. Approved in 1995 by the Food and Drug Administration for treatment of T2DM, metformin improves glucose tolerance via decreasing both hepatic production and intestinal absorption of glucose and increasing peripheral glucose uptake and utilization [100]. Metformin appears to lower fasting glucose via activation of AMP-kinase (AMPK), a pivotal molecule that redirects substrate utilization away from glucose and toward fatty acid beta oxidation. AMPK activation by metformin inhibits the mechanistic target of rapamycin (mTOR) in the liver with resultant downstream events including suppression of hepatic neogenesis [101]. Metformin crosses the blood–brain barrier and may elicit anti-inflammatory, neuroprotective effects [102,103]. In addition to the critical role of metformin in modulating metabolic and inflammation pathways that influence obesity, the drug's well-established efficacy and safety profile for T2DM and prediabetes enable the possibility of novel repurposing. Below, we review the literature on the effects of metformin on mood and cognition in both animal and human populations.

## 7. Metformin on Mood and Cognition

Several preliminary preclinical and clinical studies suggest that metformin could impact both mood and cognition. Metformin could benefit mood and cognitive functioning by (1) preventing or ameliorating metabolic dysfunction [104] and (2) acting through

cerebrovascular or neurodegenerative mechanisms [105], including decreasing advanced glycation end products [106,107], and affecting inflammation [108].

In rodent models of high-fat diet (HFD)-induced obesity, metformin is associated with amelioration of obesity-associated phenotypes suggestive of mood and cognitive dysfunction. Behavioral assessments in mice/rats include delayed learning and memory by Morris water maze test [109], behavioral despair by tail suspension test (TST) [110], anhedonia/depressive-like behavior by sucrose splash test [111], and anxiety-like behavior by elevated plus maze (EPM) [112]. In HFD-induced insulin-resistant male C57BL/6 mice, metformin leads to anxiolytic and antidepressant-like changes in behavior with increased entry/time in open spaces (EPM) and decreased time of immobility (TST) [113]. The improved behavioral effects of both HFD reversal and metformin may relate to correction of the metabolic dysfunction of diet-induced obesity and insulin resistance. Interestingly, HFD is associated with decreased basal extracellular 5-hydroxytryptamine (5-HT or serotonin) levels in the hippocampus, the region of the brain involved in emotional regulation [110]. Increased activity of 5-HT neurotransmission in the hippocampus appears to improve behavioral functioning; this effect appears mediated in part by decreasing the elevated level of branched chain amino acids (BCAAs) [113]. Notably, increased BCAA levels are associated with HFD, insulin resistance, and obesity [114]. Metformin may act to decrease circulating levels of BCCAs [115] and thus facilitate hippocampal 5-HT neurotransmission in HFD-fed mice with resultant antidepressant-like behavioral changes [113].

Metformin impedes the learning and memory behavioral decline found in insulin-resistant HFD-fed rats as a result of amelioration of metabolic abnormalities and oxidative stress levels [109]. Pretreatment with metformin also reduced learning and memory deficits in murine models of drug-induced cognitive dysfunction [116,117] via decreased inflammation [118]. In a study of rats exposed to chronic restraint stress and HFDs, both depressive symptoms and deficits in spatial memory were attenuated by metformin, fluoxetine, and combined metformin + fluoxetine; downregulation of hippocampal c-jun expression was demonstrated [119]. Cognitive decline associated with diabetes and metabolic dysfunction may in part relate to hyperglycemia-induced formation of advanced glycation end products (AGEs) and reactive oxygen species, with a recent report of hippocampal spatial memory impairment in mice with streptozocin-induced diabetes [120]. Metformin has been shown to inhibit AGE-induced proinflammatory signal transduction via AMPK activation and receptor of AGE/NF-κB, with decreased mRNA levels of proinflammatory cytokines (e.g., IL-1β, IL-6, TNF-α) in murine macrophages treated with metformin [121].

In addition to metabolic and psychological dysfunction in HFD-induced models, metformin also appears effective in reversing corticosterone-mediated metabolic dysfunction and depression-like behaviors [122,123] as well as lipopolysaccharide-induced depressive-like behaviors associated with aberrant glutamatergic neurotransmission and inflammation-related pathways [124,125]. Chronic stress is associated with elevated corticosteroid levels and prolonged activation of the sympathetic nervous system, leading to increased visceral fat and metabolic derangements that include insulin resistance and T2DM [72,89]. In a study of stress-induced behavior, C57BL/6 female mice were subjected to chronic swim testing followed by treatment with specific steroid hormone antagonists including metformin as an androgen antagonist [126]. Metformin was associated with reversal of behavior changes of increased sociability and decreased social novelty [126]. Other rodent models of depressive behavior induced by psychologic stressors, such as chronic social defeat stress (CSDS) and chronic unpredictable mild stress (CUMS), are also highly responsive to metformin treatment [127,128]. In CSDS, the antidepressant effects of metformin may involve enhanced expression of brain derived neurotrophic factor (BDNF) in hippocampal tissue/cells via AMPK activation [128]; however, in CUMS vs. control animals, BDNF protein levels in the hippocampus did not differ significantly between CUMS vs. control animals [127].

Metformin treatment of male C567B/L6 middle-aged mice via diets with 0.1% $w/w$ metformin vs. untreated mice showed improved metabolic parameters, i.e., reduced body

weight and improved metabolic profile (reduced insulin, cholesterol, and HOMA-IR) with anti-inflammatory gene expression in liver tissue [129]. In male C567B/L6 mice fed low fat 4.3% (*w/w*) vs. high fat 34% (*w/w*) for 10 weeks, mice fed high-fat diets had higher levels of fat mass, insulin, and blood glucose; adipose tissue in HFD-fed mice had higher levels of macrophages markers CD11c, MCP-1, CD206, and Arg1 [130]. An anti-inflammatory mechanism may represent the common underlying basis for improved psychological function in obesity/HFD murine models of anxiety and depression-like behaviors that involve an obese, HFD-fed phenotype.

Notably, most rodent studies of obesity and depressive-like behavior have been conducted in male mice. Future studies should address sex-specificity of the models (e.g., CSDS may only apply to male mice).

Studies exploring the effects of metformin in mood and cognition in humans have yielded more mixed, but promising, results. A recent review suggested that antihyperglycemic drugs such as metformin show efficacy in amelioration of depressive symptoms and cognitive impairment [131], suggesting an important new area of investigation. In humans, observational data from a representative cohort of 800,000 Taiwanese participants showed that metformin and sulfonylureas were associated with reduced hazards ratios for affective disorders (major and unipolar depression, and bipolar disorders) among those with T2DM [132]. At the same time, however, underdiagnoses of affective disorders is likely as the stigma is great in Asian countries [133]. Importantly, the observational nature of the data makes it challenging to draw causal conclusions. However, another case–control study of over 500 elderly patients with T2DM found that patients taking metformin had a lower risk of depression (using the Geriatric Depression Scale 1–15) than those taking no medication [134].

Metformin, after 6 months of treatment, has also been found to improve emotional functioning (as indexed by significant increases in vitality, mental health, and sum scores on the Short Form Health Survey and lower scores on the Symptom Check-List) in a study of women with polycystic ovary syndrome (PCOS) [135]. Findings from this observational study were supported by a prospective cohort study comparing patients with PCOS prescribed lifestyle modifications + metformin versus lifestyle modifications alone, which found that those in the metformin group had 70% lower risk of having major depression [136].

Guo and colleagues investigated the effects of metformin versus placebo on depression using both the Montgomery–Asberg Depression Scale and the Hamilton Scale for Depression in a sample of patients with both depression and diabetes [137]; study results showed that chronic metformin treatment over the course of 24 weeks led to significant improvement in both depression scales. Importantly, improvements in HbA1c significantly correlated with improvements in depressive symptoms, suggesting that the antidiabetic effects of metformin might mediate elevations in mood. In a pilot trial of metformin for mood disturbances in adolescent and adult women with PCOS, results demonstrated that both depression (BDI-II) and anxiety (BAI) were significantly decreased [138]. Insulin resistance and body adiposity also improved. Given the small sample size, it was not possible to determine whether changes in metabolic health explained the effects of metformin on mood. Notably, a majority of these participants with anxiety had either mild or no depression at baseline, suggesting that improvements can be evident even without high levels of depression [138].

In addition, metformin is associated with amelioration of mild cognitive impairment [139] and higher performance on cognitive tasks including memory and executive function [140,141]. These beneficial effects on cognitive function are substantiated by a few pilot clinical trials, showing that metformin has the potential to improve a range of cognitive outcomes. In the study of Guo et al., participants diagnosed with depression and T2DM demonstrated improved cognitive performance via the Wechsler Memory Scale-Revised after 24 weeks of metformin as compared to a placebo control; improvements in glucose metabolism were also evident [137].

Notably, a randomized, placebo-controlled trial of metformin in adults (55–90 years old, *n* = 80) with amnestic mild cognitive impairment and without treated diabetes showed that metformin significantly improved cognitive performance [142]. In a randomized double-blind placebo-controlled crossover study of metformin, participants (age 55–80) with mild cognitive impairment or early dementia due to Alzheimer's disease, without history of diabetes mellitus or prediabetes, showed significant improvement in cognition in the metformin group as assessed by executive functioning [143]. Improvements in learning, memory, and attention did not reach significance, possibly due to the small sample size (*n* = 20) [143]. Metformin has also been associated with enhanced declarative and working memory among survivors of pediatric brain tumors [144].

Notwithstanding, cognitive and mood effects have not always been found. In one study comparing exercise, metformin, and exercise + metformin on health-related quality of life measures in participants with T2DM, both the exercise and the exercise + metformin interventions demonstrated significant effects on mood as measured via the Profile of Mood States-SF, with large effect sizes for vigor and moderate effective sizes for anger and total mood disturbance in comparison to the metformin group alone [145]. At the same time, study limitations included an elderly patient cohort (mean age 70.6), nonrandomized design, and small sample size of the metformin-only group (*n* = 30) relative to the metformin + exercise group (*n* = 147). Although these limitations make it difficult to interpret the findings, this study suggests that metformin + lifestyle intervention can significantly improve mood. In one randomized, double-blind trial, metformin did not reduce depression scores in young women with insulin resistance due to PCOS and comorbid diagnosed major depression [146]. However, limitations to this small study included a significantly shorter treatment duration than past studies (6 weeks) and the small sample size (*n* = 25 in each group) exacerbated by a significant dropout rate (20% did not return after the first post-baseline visit). Notably, metformin did not affect the HOMA-IR (Homeostatic Model Assessment for Insulin Resistance) values, which is not surprising if the effects of metformin on depression are mediated through metabolic parameters. While an 8% reduction of depression scores from baseline was found for metformin among participants with PCOS and major depressive disorder, pioglitazone was superior with a 38.3% reduction in depression scores [146]. Data from the Diabetes Prevention Program (DPP) showed no significant group (intensive lifestyle, metformin, and placebo) differences in both depression levels as well as proportion of participants taking antidepressant medication [147].

Metformin has also been found to be associated with declines in cognitive test performance over time [148,149]. These findings, however, are observational [149] and with older adults. In one randomized clinical trial examining the effects of metformin as compared to a placebo on spatial and verbal memory in youth with autism spectrum disorder, researchers found no significant differences in memory [150]. However, these participants were taking atypical antipsychotic medications with the side effect of weight gain; in addition to potential interactions between metformin and psychotropic medications, the authors noted the severe challenges of assessing cognitive functioning in this study cohort.

Data from Diabetes Prevention Outcomes Study found no significant group differences of original randomization condition (metformin, lifestyle change, and placebo) on cognitive outcomes including verbal learning, letter fluency, and the digit symbol substitution tests [151]. However, limitations to this study include the lack of baseline cognitive assessment, collection of cognitive assessments at ~12 years post-randomization, and older age of participants (mean age = 63 years). Furthermore, differences in diabetes and glycemia among the intervention arms were also significantly smaller at the time of cognitive assessments. Despite the apparent lack of effect of metformin exposure on cognition, after adjusting for age, sex, education, and randomization arm, the Diabetes Prevention Outcomes Study results suggest a significant association between higher glycated hemoglobin and lower cognitive performance, which is consistent with data from the Finnish Diabetes Prevention Study [152].

Although prior reports from the DPP showed that weight loss is associated with a reduction in depression symptoms (regardless of randomization assignment) [147], the DPP was not intentionally designed to test the effects of metformin on mood. Eligibility criteria for the DPP allowed antidepressant medication use, which may minimize the impact of the intervention on depression levels. Moreover, at baseline, only 2.7% of participants had scores greater than 16 (moderate depressive symptoms) [99,147]. Interestingly, however, antidepressant use was associated with diabetes risk in the intensive lifestyle arm but not the metformin arm.

The effects of metformin on mood and cognition may depend on specific mechanistic targets, which may be moderated by baseline obesity levels. The cognitive benefit conferred by metformin may only occur in the context of psychological dysfunction due to metabolic brain stress [150], including T2D, hyperglycemia, and hyperinsulinemia. One study from the DPP found no significant group differences, but participants who were active or who lost weight also had reductions in depression markers regardless of treatment arm [147]. However, for men randomized to metformin, increases in total testosterone were associated with decreases in depression and anxiety [153].

Finally, the beneficial effects of metformin versus placebo on cognition (as assessed by verbal functioning) seems to be most beneficial for those with baseline BMI of 35 and above, although these effects were not significant [154]. In another study, researchers investigated not only metformin but compared rosiglitazone (an insulin sensitizer that reduces glucose levels by increasing hepatic and peripheral tissue sensitivity to insulin) to glyburide (which reduces glucose levels by enhancing insulin secretion from the pancreas) for effects on cognitive functioning among adults with diabetes but without current depression [155]. Results revealed benefits in both groups, and the magnitude of the effects were correlated with improvements in fasting plasma glucose levels but not circulating insulin or insulin sensitivity [155]. This is consistent with findings that pioglitazone effects on depression are due to mechanisms that are largely unrelated to its insulin-sensitizing action [146].

## 8. Discussion and Recommendations

Obesity is a growing and unchecked problem worldwide but particularly in the United States, where over 40% of U.S. adults have obesity and consequently increased risk for adverse health conditions such as cardiovascular disease, Type 2 diabetes mellitus (T2DM), and cancer. Compelling evidence from epidemiologic, clinical, and basic science studies indicate that the persistence of obesity is reinforced by poor metabolic health and psychologic dysfunction (depressed mood, anxiety, and cognitive impairment) in a positive feedback loop. While the obvious answer may be to treat these mood and cognitive dysregulations with antidepressants, nearly 50% of patients experience no response to treatment with first-line antidepressants [156]. Importantly, antidepressants are also often associated with significant weight gain [157]. Coupled with other potential side effects, antidepressants may be undesirable for certain populations. Finally, even though behavioral therapies appear effective at improving mood, the effort to engage in these interventions can be challenging for some populations, thus leading to adherence problems [90].

While not designed as a systematic review of the literature, this paper nonetheless draws attention to the metabolic and psychological derangements resulting from and promoting excess adiposity and high BMI as a vicious cycle from which escape is difficult if not impossible. Given the limited number of studies on metformin, we reviewed this literature thoroughly, and our paper draws on a wide range of research to make the argument for disrupting the mood–obesity cycle. While recognizing the importance of addressing the multiplicity of factors underlying the development of obesity, we wish to highlight the potential of metformin as a tolerable, safe, inexpensive intervention that may, at least in some persons with obesity, help disrupt the vicious cycle of excess fat mass, psychological dysfunction, and dysregulated metabolism.

There is compelling evidence that metformin has the potential to improve metabolic and inflammatory markers, in turn leading to better mood and cognitive functioning

and thus motivating behavior change. To date there are no well-powered randomized controlled clinical trials designed to specifically test the role of metformin on mood and cognition. At the same time, the preliminary data suggest that metformin, by acting on glucose level and inflammatory processes, may lead to improved psychological functioning. If this proves to be the case, research that tests the synergistic effect of metformin preceding a lifestyle intervention, or along with a lifestyle intervention, may be a transformative pathway by which clinicians can help motivate behavior change in individuals with obesity. By addressing mood dysregulation and cognitive deficits that make it challenging to initiate and sustain healthy eating and exercise, metformin could potentially act as a jumpstart for individuals for whom initiation of lifestyle changes may be intractable. Additionally, because metformin has an established significant impact on metabolic health, and may also improve mood and cognition, pairing metformin with a lifestyle intervention could potentially lead to an early response (e.g., behavior change). Evidence suggests that individuals who exhibit early behavioral change [158] or greater early weight loss [159] are more likely to be successful in long-term weight loss and maintenance.

**Author Contributions:** Conceptualization, writing—original draft preparation, visualization: S.N.D. and L.D.Y.; writing—review and editing, S.K.P., B.X. and R.A.N. All authors have read and agreed to the published version of the manuscript.

**Funding:** This research received no external funding.

**Institutional Review Board Statement:** Not Applicable.

**Informed Consent Statement:** Not Applicable.

**Data Availability Statement:** Not Applicable.

**Conflicts of Interest:** The authors declare no conflict of interest.

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
