# Peer review of "Disrupting the Mood and Obesity Cycle: The Potential Role of Metformin"

_2673-4168, doi:10.3390/obesities3010006_

Round 1
Reviewer 1 Report
Comments and Suggestions for Authors
The authors present an interesting perspective on the potential role of Metformin in mitigating mood disruptions related to obesity. As a perspective piece, there is a lot to like, as the paper is well written and in some ways convincing. That said, it's not really a review in the modern sense of the word, as it lacks a systematic process to select and evaluate the literature. Rather, the authors select literature that supports their central thesis without critically appraising it. I would suggest that the authors either back up a step and conduct a systematic review (or two...there is a lot to cover in this manuscript) or submit this manuscript as a perspective article at a journal that accepts that article type. If they choose to conduct a systematic review, they should be very critical of the literature reporting relationships between obesity and mood disorders that don't also assess physical activity, diet, and sedentary time as mediators of the relationships. Finally, the authors should revise the manuscript to use person first language throughout (i.e., persons with obesity rather than obese persons).
Author Response
If they choose to conduct a systematic review, they should be very critical of the literature reporting relationships between obesity and mood disorders that don't also assess physical activity, diet, and sedentary time as mediators of the relationships.
We revised the Discussion to address this limitation:
While not designed as a systematic review of the literature, this paper nonetheless draws attention to the metabolic and psychological derangements resulting from and promoting excess adiposity and high BMI as a vicious cycle from which escape is difficult if not impossible. Given the limited number of studies on metformin, we have reviewed this literature thoroughly, and our paper draws on a wide range of research to make the argument for disrupting the mood-obesity cycle. While recognizing the importance of addressing the multiplicity of factors underlying the development of obesity, we wish to highlight the potential of metformin as a tolerable, safe, inexpensive intervention that may at least in some persons with obesity help disrupt the vicious cycle of excess fat mass, psychological dysfunction, and dysregulated metabolism.
the authors should revise the manuscript to use person first language throughout (i.e., persons with obesity rather than obese persons)
We revised the document to use person first language, e.g., replacing ‘obese persons’ with ‘persons who are obese.’
Reviewer 2 Report
Comments and Suggestions for Authors
Dear Authors, this is an interesting topic and you have provided a good review about one of the major helath issues nowadays.
In introduction, could you please kindly provide more details about the reasons that you have chosen to include only women in your paper?
You have stated that obesity and overweight status are closely associated with mood dysregulation in a 78 co-dependent, bidirectional manner - could you please provide more information about the role of obesity on depression development?
I suppose Figure 1 was made by one of the authors? Please state it below the Figure.
You have provided a detailed explanation about the impact of metformin, but could you explain why you have included only metformin? I suppose the reason is that other medications do not act through AMPK activation.
Reference list is not according to the Journal's reference style recommendations.
Author Response
In introduction, could you please kindly provide more details about the reasons that you have chosen to include only women in your paper?
We removed the initial framing of the paper around women as the review overall incorporates extant research inclusive of both men and women: 1) deleted sentence at lines 41 to 42, This is particularly true for women - only 19% of overweight and 16% of obese women meet the minimum recommended amount of activity; 2) deleted “women” line 43; and 3) changed “women” to “people” line 48.
You have stated that obesity and overweight status are closely associated with mood dysregulation in a 78 co-dependent, bidirectional manner - could you please provide more information about the role of obesity on depression development?
We added this paragraph at line 231-242, with 2 additional references:
Obesity is also linked to mood dysregulation as a risk factor. Chronic inflammation arising from higher fat mass and metabolic dysfunction appears associated with psychological effects such as depression and anxiety [Mann et al, 1999]. In a study of data from a mental health questionnaire of participants in the U.K. Biobank (n=145,668), Casanova et al demonstrated that higher adiposity leads to higher odds of depression, severity of depression, and lower well-being regardless of genetic predisposition to metabolic dysfunction (e.g., adiposity genetic variants with favorable or unfavorable metabolic profiles based on HDL cholesterol, triglycerides, and T2DM risk) [Casanova et al, 2021]; limitations of the analysis include lack of diversity in the European study population and potential bias in the subset of partipants involved in the mental health questionnaire substudy. Interestingly, the metabolically favorable adiposity variants were associated with higher levels of the proinflammatory cytokine CRP [Casanova et al, 2021].
I suppose Figure 1 was made by one of the authors? Please state it below the Figure.
“figure designed by LDY, SD” is now included.
You have provided a detailed explanation about the impact of metformin, but could you explain why you have included only metformin? I suppose the reason is that other medications do not act through AMPK activation.
We have added a sentence to clarify this point:
In addition to the critical role of metformin in modulating metabolic and inflammation pathways that influence obesity, the drug’s well-established efficacy and safety profile for T2DM and prediabetes enable the possibility of novel repurposing.
Reference list is not according to the Journal's reference style recommendations.
We reformatted the reference list to the Journal’s reference style recommendations (ACS format).
Reviewer 3 Report
Comments and Suggestions for Authors
The manuscript “Disrupting the mood and obesity cycle: the potential role of 2 metformin” is a timely review on the literature on putative effects of metformin to blunt progression of a cycle where disruption of mood and cognitive function by metabolic dysfunction in overweight and obese individuals further promotes weight gain. The review puts forward a testable possibility, namely to use metformin preceding, or along with a lifestyle intervention to help reduce weight. However, the review is difficult to read because, while it is apparently meant to include a general audience interested in metabolism and weight gain, it then uses specialized terms that are not well explained.
· Line 97. “Higher negative affect [34] and lower positive affect [35], as assessed using the Positive and Negative Affect Schedule, are associated with increased BMI; for women, the effect is stronger 99 [35]”
Please define negative and positive affect
· Line 101. “Mood disorders like depression are thought to lead to excessive weight gain because depressed individuals tend to have lowered energy and therefore are less physically active and more inclined to eat foods that require less preparation effort”
Please add relevant references, I think they are later in the text, but must be added here too.
· Line 106. “Moreover, analyses suggested that depression was more likely to precede obesity, rather than vice versa.”
Please add relevant references, same as above.
· Line 109 “and the intransigent cycle of mood and metabolic dysfunction [39, 40”].
Please revise wording on “intransigent cycle”, what does that mean?
· Line 114 “including working memory, inhibitory control, and shifting/flexibility [42], as well as decision-making [43], verbal fluency [44], and planning [45].
Define parameters also in light of the following sentence “Specifically, obese individuals exhibited poor EF across all domains, while overweight individuals only showed deficits in inhibition and working memory [41]. What is “inhibition”, what is “working memory”?
· Line 140 define WHR
· Line 170. “These data suggest that obesity is a “disorder of appetitive motivation,” rather than simply disruptions in homeostatic mechanisms of food intake”.
This sentence ignores the role of melanocortin system in eating behavior.
· Line 182 “depressed mood/negative affect leads to an increase preference for high-caloric food, which can serve as a form of emotion regulation”
Please add reference
· Line 466 “Compelling evidence from epidemiologic, clinical, and basic science studies indicate that the intransigence of obesity is reinforced by poor metabolic”
Please change intransigence with another word or change the sentence
Author Response
Please define negative and positive affect
We added the bold, italicized text (lines 100-102):
Line 97. “Higher negative affect or emotional distress (e.g., anxiety, depression, stress, sadness) [34] and lower positive affect or pleasant feelings or emotions (e.g., joy, calmness, interest, enthusiasm) [35], as assessed using the Positive and Negative Affect Schedule, are associated with increased BMI; for women, the effect is stronger [35]”
Please add relevant references.
Lines 105-108 of revised draft:
Line 101. “Mood disorders like depression are thought to lead to excessive weight gain because depressed individuals tend to have lowered energy and therefore are less physically active [Roshanaei-Moghaddam et al, 2009] and more inclined to eat foods that require less preparation effort negative affect tends to be associated with higher intake of sweet, high-fat, and more energy dense foods [Oliver et al, 2000].”
Line 112 of revised draft:
Line 106. “Moreover, analyses suggested that depression was more likely to precede obesity, rather than vice versa [Fabricatore et al, 2004].”
Please revise wording on “intransigent cycle”, what does that mean?
We used inexorable in place of intransigent on line 114-115:
Line 109 “and the intransigent inexorable cycle of mood and metabolic dysfunction [39, 40”].
Define parameters also in light of the following sentence: “Specifically, obese individuals exhibited poor EF across all domains, while overweight individuals only showed deficits ininhibition and working memory [41].
Lines 120-126 were revised to include explanations for the components of executive function in bold, italicized font:
Line 114 “including working memory (i.e., short term storage of relevant, immediate information), inhibitory control (i.e., ability to control one’s attention, behavior, thoughts, and/or emotions to override impulsive or automatic/conditioned responses), and shifting/flexibility (i.e., ability to adapt behavior and thoughts to new, changing, or unexpected events) [42], as well as decision-making, which include elements of applying and following rules [43], verbal fluency (a measure of speed of semantic processing) [44], and planning, or forethought for future adaptive responses [45].
What is “inhibition”, what is “working memory”?
We defined the terms as detailed above. We changed inhibition as a more colloquial term to “inhibitory control as used above.
“Specifically, obese individuals exhibited poor EF across all domains, while overweight individuals only showed deficits in inhibition inhibitory control and working memory [41].
Line 140 define WHR
Waist Hip Ratio or WHR as ratio of measurements of waist to hip circumferences is included on Line 153.
This sentence ignores the role of melanocortin system in eating behavior.
We added the sentence in bold and italics with two new references (Lines 185-187) :
Line 170. “These data suggest that obesity is a “disorder of appetitive motivation,” rather than simply a disorder of disruptions in homeostatic mechanisms of food intake”. Dysfunction of central melanocortin system, which is involved in regulation of energy homeostasis, food intake, satiety, and body weight, is also implicated in the pathogenesis of obesity [Micioni et al 2020, Goit et al, 2022].
Please add reference:
Reference has been added to Line 199.
Line 182 “depressed mood/negative affect leads to an increase preference for high-caloric food, which can serve as a form of emotion regulation [Singh 2014].
Please change intransigence with another word or change the sentence.
We changed intransigence to persistence on Line 499.
Line 466 “Compelling evidence from epidemiologic, clinical, and basic science studies indicate that the intransigence persistence of obesity is reinforced by poor metabolic”
Round 2
Reviewer 1 Report
Comments and Suggestions for Authors
I appreciate the author's attempt to use person first language throughout, but replacing ‘obese persons’ with ‘persons who are obese' doesn't correct the problem. The point is that obesity is a disease. You wouldn't say 'persons who are cancer' so you shouldn't say ‘persons who are obese'. Please correct to say 'persons with obesity' or something similar.
Round 3
Reviewer 1 Report
Comments and Suggestions for Authors
Thank you for making the requested changes.